# Interactome of Long Non-Coding RNAs: Transcriptomic Expression Patterns and Shaping Cancer Cell Phenotypes

**DOI:** 10.3390/ijms24129914

**Published:** 2023-06-08

**Authors:** Nicole R. DeSouza, Danielle Quaranto, Michelle Carnazza, Tara Jarboe, Raj K. Tiwari, Jan Geliebter

**Affiliations:** 1Department of Pathology, Microbiology and Immunology, New York Medical College, Valhalla, NY 10595, USA; 2Department of Otolaryngology, New York Medical College, Valhalla, NY 10591, USA

**Keywords:** noncoding RNA, microRNA, piRNA, lncRNA, circRNA, disease pathogenesis, developmental defects, cancer

## Abstract

RNA biology has gained extensive recognition in the last two decades due to the identification of novel transcriptomic elements and molecular functions. Cancer arises, in part, due to the accumulation of mutations that greatly contribute to genomic instability. However, the identification of differential gene expression patterns of wild-type loci has exceeded the boundaries of mutational study and has significantly contributed to the identification of molecular mechanisms that drive carcinogenic transformation. Non-coding RNA molecules have provided a novel avenue of exploration, providing additional routes for evaluating genomic and epigenomic regulation. Of particular focus, long non-coding RNA molecule expression has been demonstrated to govern and direct cellular activity, thus evidencing a correlation between aberrant long non-coding RNA expression and the pathological transformation of cells. lncRNA classification, structure, function, and therapeutic utilization have expanded cancer studies and molecular targeting, and understanding the lncRNA interactome aids in defining the unique transcriptomic signatures of cancer cell phenotypes.

## 1. Introduction

RNA biology has emerged in the last two decades as a key target for both diagnostic and therapeutic intervention. The versatility of RNA molecules makes them attractive and dynamic biochemical targets for study; elucidation of RNA mechanisms gives great insight as to how cells utilize the genome and expend cellular energy. The long-considered central dogma of biology follows the schematic that DNA is transcribed into RNA, and RNA is subsequently translated into a protein product. The coding portion of the transcriptome only accounts for 2%, leaving the remaining 98% referred to as the “dark matter” of the genome, or “junk” DNA [1]. These transcripts were thought to be non-functional by-products of RNA polymerase II transcription and completely lack biological function due to the absence of open reading frames [2]. However, in recent years, indebted to Next Generation Sequencing technology, the genomic transcripts that do not encode for functional protein products have gained the spotlight for their differential expression patterns in a multitude of pathological conditions. These noncoding transcripts are referred to as “noncoding RNAs” (ncRNAs) and have been identified as critical molecular regulators and adjustors of genetic material through transcriptional, post-transcriptional, and post-translational modifications.

Of particular focus, long noncoding RNAs (lncRNAs) mediate a plethora of molecular and cellular processes. Physiologically, lncRNA molecule expression has been demonstrated to govern and direct cellular differentiation and transcriptional regulation. Thus, it is evident that instances of aberrant lncRNA expression can drive the pathological transformation of cells. Often, lncRNA molecules become relevant in differential gene expression studies of pathological conditions, such as cancer. Both the over- and under-expression of lncRNAs in cancerous vs. noncancerous tissue provide novel avenues for evaluation of the epigenetic programs cancer cells employ to promote their establishment and progression. lncRNA molecules are involved in cellular processes such as proliferation, apoptosis, invasion, and migration, therefore further delineating their pathological role in instances of dysregulated expression in human disease, especially cancer (Figure 1). Thus, lncRNAs have enabled, in part, the identification of carcinogenic molecular patterns that arise in transformed cells.

## 2. lncRNA Classification

The two broadest categories of ncRNAs are divided into housekeeping and regulatory molecules. Housekeeping ncRNAs are well studied and include molecules such as transfer RNAs (tRNAs) and ribosomal RNAs (rRNAs). Regulatory RNAs are classified most broadly by their transcript size. Noncoding transcripts less than 200 nucleotides (nt) in length are denoted as “small ncRNAs”. Small ncRNAs include, but are not limited to, microRNAs (miRNAs), short interfering RNAs (siRNAs), and piwi-interacting RNAs (piRNAs). Noncoding transcripts greater than 200 nt in length are classified as the aforementioned lncRNAs. All ncRNA molecules exert their biological roles as functional transcriptomic elements, lacking the ability for translation into protein products. However, despite being represented by the vast majority of the genome, individual lncRNA molecules are expressed in tissues at lower levels when compared to mRNA molecules, making them more difficult to discover and annotate [3]. 

lncRNAs are further subdivided based on their genomic locus and structure. Some examples of lncRNA subdivisions include long intergenic noncoding RNAs (lincRNAs), pseudogene-derived lncRNAs (i.e., transcribed pseudogenes), competing endogenous ncRNAs (ceRNAs), natural antisense transcripts (NATs), promoter-associated RNAs (PARs), and enhancer RNAs (eRNAs). lncRNAs can either be linear or circular (circRNA) in structure, the latter being more stable. circRNAs are composed of a loop of non-coding RNAs that have joined 3′ and 5′ ends, and these molecules are resistant to enzymatic degradation that is facilitated by endonucleases due to the lack of free 3′ and/or 5′ substrates that these enzymes require [4]. Divisions of lncRNAs vary and may overlap in categories of nomenclature.

## 3. lncRNA Structure

The most important aspect of lncRNA studies is elucidating their mechanisms of action. The ability of lncRNA molecules to form secondary structures enables their interactions with nucleic acids and proteins within the cell [5]. Unlike other nucleic acid macromolecules, lncRNAs are less likely to conserve their primary structure (i.e., unique sequence of nucleic acids) but are rather governed by the secondary and/or tertiary structures they are capable of forming [3]. The lack of sequence constraints and conservation creates an additional barrier when studying the functions of these molecules across species. Structurally, these molecules can also serve as piRNA and miRNA precursors. lncRNAs may be post-transcriptionally modified; some lncRNAs are 5′-capped, polyadenylated, and/or contain splice variants, which are modifications shared by their protein-coding mRNA counterparts [6]. The structural composition and conformation of these lncRNA molecules heavily govern the type of function they will exert within the cell.

## 4. lncRNA Function

lncRNA molecules have a diverse set of intracellular capabilities and functions. The area of cellular residence gives significant insight into the molecular components they will interact with. For example, lncRNAs can act in the cis or trans regions of DNA as well as with RNA and proteins. Cis-regulatory lncRNAs typically target their “genomic neighborhood”, i.e., genes present on the same chromosome as their own genomic locus and are nuclear residents. Trans-regulatory lncRNAs modulate gene expression outside of their own chromosomal location; these lncRNAs can reside in the nucleus, mitochondria, cytoplasm, or be exported as exosomal cargo. lncRNAs have been mainly identified to function as signals, signal decoys, protein and molecular scaffolds, and negative regulators of miRNAs. These varying functions are a direct result of functional RNA-binding domains, DNA-binding domains, and protein-binding domains present within the lncRNA structure, greatly influencing the versatility and complexity of their biological action (Figure 2). As such, lncRNA molecules can orchestrate both physiological and pathological processes, depending on their expression levels and localization.

lncRNA molecules are identified as having tissue-specific expression patterns and the ability to respond to different stimuli. For example, the Xist (X-inactive-specific-transcript) gene encodes a lncRNA involved in X chromosome inactivation and functions to silence gene expression on the inactive X chromosome during female embryonic development [7]. Despite its known physiological role, both increased and decreased expression of Xist have been pathologically correlated with several tumor types. In glioblastoma, Xist upregulation supports tumor progression through a sponging interaction (see below) with miR-152 [8], and in nasopharyngeal carcinoma, Xist has an oncogenic role through a sponging interaction (see below) with miR-34a-5p [9]. Alternatively, in some female-dominated cancers, such as breast cancer (BC), Xist downregulation has been correlated with a worse prognosis. In BC, Xist causes inactivation of PHLPP1 (PH Domain and Protein Rich Phosphatase) through epigenetic signature alteration, resulting in an upregulation of Akt activation, a central protein of proliferative signal transduction pathways [10]. Xist, therefore, is a prime example of how a single lncRNA molecule can vary in expression and function in different tumor tissues.

## 5. Resources for Studying lncRNAs

Despite mutational burden being a critical driver of carcinogenesis, cancer can arise from a multitude of epigenetic processes that fine-tune wild-type genomic loci. lncRNAs have been identified for their differential expression patterns in cancerous tissue, which is directly correlated to cellular pathways and processes that heavily promote central metastatic mechanisms [11]. Additionally, a single lncRNA can have vastly diverse functions in different cancer types. The need for tumor biomarkers is pertinent; these markers should be easily detectable in body samples and uniform across cancer types, contributing immensely to rapid detection and prognostic value [12]. Preliminary bioinformatic analyses of patient tissue gene expression levels have enabled the identification of novel biomarkers and therapeutic targets. Profiling through fold-change analysis for pairwise gene comparison has been useful for identifying profiles that are among the most significantly differentially expressed in diseased tissue [13]. Publicly available databases provided by the NCBI (National Center for Biotechnology Information) and the GEO (Gene Expression Omnibus) enable the analysis of a reservoir of patient data and gene profiling using GEO2R software [14]. GEO2R software (accessed on 1 May 2023, https://www.ncbi.nlm.nih.gov/geo/) combines GEOQuery and limma (Linear Models for Microarray Analysis) R packages provided by the Bioconductor project for the successful analysis of high-throughput genomic data. The Cancer Genome Atlas (TCGA) is a database that contains 2.5 petabytes of genomic, transcriptomic, epigenomic, and proteomic data from over 20,000 primary cancer and matched normal patient samples across 33 different cancer types (accessed on 1 May 2023, https://www.cancer.gov/tcga). The cBioPortal for Cancer Genomics was developed at Memorial Sloan Kettering Cancer Center and is a public portal for cancer genomic data from patient samples. The cBioPortal contains putative DNA copy numbers, mRNA and miRNA expression data, as well as epigenetic data such as DNA methylation patterns [15,16]. Gene Expression Profiling Interactive Analysis (GEPIA) enables the analysis of RNA sequencing data from almost 10,000 tumor samples and 9000 normal samples provided by the TCGA and GTEx (Genotype-Tissue Expression) projects. The GTEx database enables the molecular analysis of gene expression profiles that are tissue-specific. This is an invaluable tool for ncRNA study, as ncRNA molecule expression is tissue-specific and tightly controlled. GTEx contains samples from 54 healthy tissue sites that contain genetic analysis using RNA-seq and whole-genome sequencing (WGS) data (accessed on 1 May 2023, https://gtexportal.org/home/). GEPIA is a useful tool for analyzing differential gene expression patterns, but it also has access to survival data and the impact of genomic profiles on patient survival. This tool can be used for in silico prediction of interactions between two genomic (or transcriptomic) profiles based on their correlation coefficient, as well as identification of the effect of a specific genomic profile on survival in a specific cancer type [17]. NetworkAnalyst 3.0 is a software tool that enables comprehensive gene expression profiling as well as the identification of differentially expressed genes within datasets [18]. Once candidate lncRNAs have been identified through these gene expression analyses, assessing their functional roles gives tremendous insight into the molecular profile of the cancer type being studied.

## 6. lncRNAs as “Molecular Sponges”

The RNA-binding domains of lncRNAs enable their action as molecular decoys, which coincides with their ability to serve as negative regulators of miRNAs. miRNAs are initially produced as primary transcripts (pri-miRNA), which are then processed into precursors (pre-miRNA), and then mature miRNA molecules. This maturation process is facilitated by RNase III DROSHA-DGCR8, which catalyzes the conversion from a pri-miRNA to a pre-miRNA, followed by the enzymatic action of RNase III DICER, which generates the final, mature miRNA transcript that contains a 5′ terminal phosphate and a 3′ hydroxyl group (Figure 3) [19]. miRNAs exert their biological action by binding to mRNAs at the 3′ untranslated region (3′UTR) of the mRNA via miRNA response elements (MREs), inhibiting mRNA translation. miRNAs function through the RNA-induced silencing complex (RISC), which is composed of Argonaute (Ago) proteins that effectively silence target mRNA transcripts. To add to this notion, lncRNAs also indirectly affect mRNA translation. This epigenetic axis sheds an additional layer of regulation where lncRNAs binding these miRNAs, referenced commonly as ceRNAs, directly impact the availability of miRNAs. In order for a lncRNA to function as a ceRNA, it requires the presence of an MRE with “incomplete complementarity” to the miRNA it is targeting. This concept is important because it highlights the fact that lncRNA–miRNA interaction does not cause rapid decay of the miRNA molecule but results in the occupation of binding sites, preventing miRNA–mRNA interaction (Figure 4) [20].

The cellular processes that are affected by this interactive epigenetic axis determine the degree to which a lncRNA molecule can govern cellular behavior, meaning that the miRNA and subsequent mRNA targets will determine the outcome that a lncRNA molecule has on the expression of a specific protein-coding gene. For example, lncRNA MALAT1 (metastasis associated with lung adenocarcinoma transcript 1), has been identified to directly target miR-9, and miR-9 is suggested to downregulate NF-κB [21,22]. Thus, MALAT1 serves as a positive regulator of NF-κB. Additionally, MALAT1 has been found to serve as a negative regulator of miR-183 in melanoma cells, thereby controlling the output expression of ITGB1 (integrin subunit beta 1). ITGB1 enhances extracellular matrix degradation as well as the migratory propensity of transformed cells [23,24]. MALAT1 also serves as a negative regulator of miR034a, which, in turn, regulates the expression of c-myc in melanoma cells, a well-known oncogene that is constitutively expressed in a multitude of malignant cancer types. This interactive cross-talk reveals MALAT1 as a critical cell cycle regulator in melanoma cells, and its overexpression has been correlated with the progression of melanoma cell growth [25]. These examples put into perspective the ability for a single lncRNA molecule to bind multiple miRNAs, in multiple cell types, thus exhibiting diversified functional capabilities.

Studying these interactions has led to the extensive development of software tools that enable in silico analysis and prediction of interacting molecules within the cell. These software tools utilize different algorithms, based on various RNA sequence parameters and literature documentation, to predict whether two RNA molecules will interact. Some examples include LncMirNET [26], LncRNA2Target [27,28], LncTar [29], IntaRNA [30,31,32,33], and DIANA-LncBase v3 [34]. Evaluation of these predictive algorithms in vitro can be accomplished using RNA-fluorescence in situ hybridization (FISH) and confocal microscopy, which would identify the localization and interaction between individual and cognate RNA molecules within a cell or tissue. Once target miRNAs have been identified, additional bioinformatic software can be implemented to identify target mRNAs. Some examples include miRnet [35,36,37,38,39] and miRTarBase [40]. Additionally, dbDEMC (database of Differentially Expressed miRNAs in human cancers) enables the identification of differentially expressed miRNAs in cancer types that are detected by high- and low-throughput techniques. This database is a meta-analysis repository of data from GEO, SRA (Sequence Read Archive), and TCGA [41]. The in silico predictions of targeted interactions between lncRNAs and miRNAs have enabled evaluation both in vitro and in vivo.

## 7. lncRNAs as Molecular Scaffolds

The protein-binding domains of lncRNAs open a multitude of actions within the cell. This enables the lncRNAs to directly impact protein localization within the cell and may promote protein interactions that would otherwise not take place in the absence of the lncRNA molecule.

Studying interactions between ncRNAs and proteins in vitro enables additional classification of transcriptomic mechanisms, as well as the spatial organization of cellular constituents. Methods for studying these interactions can either be classified as RNA-centric or protein-centric. RNA-centric analysis involves the identification of proteins that are bound to an RNA molecule of interest. Protein-centric analysis involves the investigation of the RNA molecules that are bound to a protein of interest [42]. With a focus on RNA-centric methods, an effective approach to identifying proteins bound to a lncRNA of interest involves end-biotinylation of an RNA molecule, followed by subsequent incubation with streptavidin beads. Protein fractions are then added to the column, forming cognate RNA–protein complexes. The proteins bound to the RNA can then be eluted out and identified through mass spectrometry and Western blotting. This technique is known as an RNA-pulldown method and has been successful in identifying the proteins that lncRNAs are scaffolding within the cell. A similar approach involves the use of aptamer-tagged RNAs, in which a target RNA molecule is tagged and bound to a resin support, allowing for the binding of proteins within the added cell lysate. This is a technique referred to as aptamer-tagged RNA capture [43]. Protein-centric mechanisms can also lead to the identification of novel lncRNA transcripts as well as the unknown roles of known lncRNA transcripts. A common method used to study the binding of RNA molecules to a protein of interest is the RNA Immunoprecipitation (RIP) assay. The RIP assay can enable the mapping of binding sites when the cross-linked method is used. RIP-Sequencing (RIP-Seq) enables this mapping through the use of cDNA technology [44].

Modification of chromatin by histone methylation is a crucial regulatory process that drives and harnesses transcriptional control. An important example of RNA–protein interaction includes lncRNAs that bind and direct chromatin-modifying proteins. This function enables lncRNA molecules to exert epigenetic regulation of genomic loci. lncRNAs have been identified to play varying roles in these modification processes. An example would be the action of HOTTIP (HOXA transcript at the distal tip), which exerts cis gene activation at the HOXA locus via interaction with WDR5 (WD Repeat Domain 5), promoting fibroblast differentiation [45]. Additionally, HOTAIR (HOX transcript antisense RNA) scaffolds the proteins PRC2 (polycomb repressive complex 2) and LSD1 (lysine-specific demethylase 1), which promotes H3K27 methylation and M3K4 demethylation, leading to the silencing of this genomic locus. This epigenetic silencing has been shown to promote carcinogenesis and multidrug resistance in small-cell lung cancer (SCLC). Additionally, HOTAIR promotes H3K23 methylation through direct interaction with EZH2, a protein that facilitates this methylation pattern. This interaction is shown to be directly related to SCLC cell invasion and metastasis [45]. Moreover, H3K4 methylation and H3K27 methylation regulate SLC47A2 (solute carrier family 47 member 2) expression, and the acetylation of H3K27 at the SLC47A2 promoter is required for SLC47A2 expression. SLC4A2 functions in the kidney as a transporter that is responsible for the excretion of toxic electrolyte components. Low levels of lncRNA SANT1 result in the reduction of H3K27 acetylation, a modification critical for transcription of the entire coding genomic locus. Low SANT1 levels result in increased binding of the inhibitory complex, E2F/HDAC1 (Histone deacetylase 1), which, therefore, functions as a cis-regulator of SLC47A2. Thus, low levels of SANT1 have been reported as a potential prognostic factor for renal cell carcinoma [46].

It is well known that NF-κB (nuclear factor-kB) serves as a pro-inflammatory transcriptional activator and plays central roles in carcinogenic mechanisms. A study reported that the novel lncRNA, Uc003xsl.1, binds directly to NKRF (NF-κB repressing factor), a nuclear transcription factor that tightly regulates NF-κB activity. Uc003xsl.1 inhibits the binding of NKRF to the IL-8 promoter—a critical response gene and readout of NF-κB activation. Uc003xsl.1 was identified as a highly expressed transcript in triple-negative BC (TNBC), and increased expression was correlated with poorer patient outcomes. The aberrant activation of NF-κB/IL-8 activity is reported to drive TNBC progression; thus, Uc00xsl.1 has been identified as a potential therapeutic target [47].

As evidenced, these extensive protein-binding domains make lncRNAs a central post-translational regulatory factor.

## 8. lncRNA DNA-Binding Domains

lncRNA molecules have extensive DNA-binding domains that enable the formation of structures known as “RNA-DNA triplexes”. These triplex structures have been reported to be involved in the targeting of DNA sequences in a highly specific fashion, promoting the onset and progression of carcinogenesis. RNA-DNA triplex structures can be predicted using LongTarget [48], a tool that employs base-pairing predictive rules and binding motifs present within the nucleic acid structures. In vivo methods such as ChIRP-seq (Chromatin Isolation by RNA Purification) can be employed to detect genomic regions bound by ncRNA molecules [49].

The lncRNA molecule MEG3 (maternally expressed 3) forms this triplex structure when bound to the PRC2 complex via GA-rich sequences. The formation of this triplex structure has been identified to regulate genes within the TGF (transforming growth factor)-beta pathway, and the presence of MEG3 is indispensable for activation of primary gene targets within this pathway [50]. A study reported an analysis of publicly available single-cell RNA-sequencing data and found that MEG3 is primarily expressed by cancer-associated fibroblasts (CAFs) in papillary thyroid cancer (PTC) and that MEG3 is positively correlated with lymph node (LN) metastasis [51]. It was further reported that knockdown of MEG3 in fibroblasts was correlated with a decrease in key matrix metalloproteinase expression (i.e., MMP-1, MMP-9, and MMP-16), thereby providing evidence of a role for MEG3 in tumor neovascularization and metastasis [52,53,54]. lncRNA MIR100HG (microRNA host gene) has been reported to regulate p27 transcription, a critical cell cycle control protein, through the formation of an RNA-DNA triplex, and to be highly expressed in triple-negative BC (TNBC), with its expression correlated with poor prognosis [55].

## 9. lncRNAs Impact Cellular Differentiation

lncRNA molecules can impact the differentiation and survival processes of cancer cells within the tumor environment. lncRNA-HAL is hypoxia-induced and was reported to be an overexpressed transcript in p27-positive quiescent cell populations, such as the estrogen receptor-positive (ER-pos) BC cell line MCF7. This molecule was additionally reported to promote cancer cell survival in the aforementioned hypoxic conditions, further suggesting the gain-of-function, pro-tumorigenic orchestration of up- and down-regulated genes in cancer cells [56]. Another hypoxia-induced lncRNA molecule, lncRNA NDRG-OT1 (N-Myc Downstream Regulated Gene 1-Overlapping 1), inhibits expression of NDRG1 in BC cells. Functionally, NRD-OT1 promotes the ubiquitination and thus degradation of NRDG1, which is a protein that is responsible for cellular differentiation and metastasis in BC [57].

Additional functions of lncRNAs include their regulatory roles in immune system function. The lncRNA lnc-DC has been identified as a regulator of dendritic cell (DC) differentiation through interaction with STAT3 (signal transducer and activator of transcription 3). lnc-DC binds directly to STAT3 within the cytoplasm of DCs through a stable 3‣-end stem loop structure, inhibiting its function. In the absence of lnc-DC, STAT3 binds to the SHP1 (src homology 2 domain-containing protein tyrosine phosphatase 1) protein, a known negative regulator of immune cell activation and differentiation [58]. Thus, lnc-DC functions as a positive regulator of immune function, which can dramatically impact tumorigenic establishment.

The tumor microenvironment (TME) plays a critical role in the establishment and progression of tumors. The TME of each cancer type is defined by the immune cell population and the cytokine milieu that either drives or suppresses tumor progression. The balance between anti-inflammatory and pro-inflammatory tumor infiltrates within the TMEs determines the fate of the growing tumor. lncRNA molecules have recently been identified to contribute to immunomodulatory action within the tumor realm. T helper 17 (Th17) cells are a subtype of CD4^+^ T cells that have been correlated with tumor-promoting behavior due to their production of IL-17. This mechanism has been studied in colorectal cancer (CRC), suggesting that IL-17 drives the development and progression of this cancer type in situ. There have been several other lncRNA molecules highlighted in the literature that are correlated with Th17 cell differentiation, such as NEAT1, MEG3, and H19. These lncRNA molecules and their differential expression patterns have been linked as drivers of other inflammatory diseases such as rheumatoid arthritis and endometriosis via epigenetic modulation [59,60].

As mentioned earlier, lncRNA molecules can also be present within exosomes and thus can be secreted from cells and impact the extracellular environment. Tumor cells also secrete exosomes, and their exosomal cargo greatly influences their survival and communication within the TME. Sun et al. provided evidence that lncRNA CRNDE h (Colorectal neoplasia differentially expressed; h: exosomal cargo isoform), present in CRC cell exosomal cargo, promotes the metastatic propensity of CRC. CRNDE h was found to promote a Th17 phenotype through the induction of RORγt expression, which in turn, drives IL-17 activation and is a key marker of a Th17 phenotype. CRNDE h functions through direct binding of RORγt, which inhibits the binding of the protein Itch, thus preventing itch degradation. Preventing this degradation leads to a favored Th17 and Treg (T regulatory) (see below) phenotype via Itch-mediated mechanisms [61].

In addition to Th17 cells, the immune suppressive environment that promotes tumorigenic establishment is also driven by another subtype of CD4^+^ T cells, Treg cells. Tregs produce and release anti-inflammatory cytokines that shut down (or dampen) the immune response, enabling the tumor to evade immune system recognition. IDO (indoleamine 2,3-dioxygenase), an inducible enzyme, contributes to immune suppression by promoting Treg differentiation and maturation, and is, in turn, negatively regulated by miR-448. The lncRNA molecule SNHG1 (small nucleolar RNA hostgene 1) was found to directly bind and “sponge” miR-448, thereby increasing IDO expression. IDO has additionally been found to promote the immune escape of malignant tumors, protecting them from immune surveillance [62]. Thus, increased SNHG1 expression is subsequently correlated with increased IDO expression through the lncRNA-miRNA-mRNA regulatory axis. Pei et al. conducted in vivo experiments that confirmed that high SNHG1 expression in their BC model contributes to the immune escape of malignant BC at the aforementioned interactive levels of miR-448 and IDO [63].

## 10. Clinical Relevance of RNA Molecules

The identification of how ncRNAs function to direct genomic expression has given way to the development of novel therapeutics that can harness these functions, targeting genes that lie at the hub of disease progression. Additionally, the identification of ncRNAs has given insight into other molecules that can be targeted outside of the coding genome and can still impact the expression levels of genes. Thus, the understanding of the biological relevance of noncoding RNA molecules, as well as other transcriptomic elements, has opened up novel avenues for prognostic, diagnostic, and therapeutic intervention.

## 11. RNA Molecules as Prognostic Factors

The various roles of lncRNAs discussed in this paper support the utilization of these molecules as prognostic factors. Identifying and understanding the roles of lncRNAs that are differentially expressed is a vital prerequisite for determining prognostic value. Specifically, lncRNA roles as “molecular sponges” enables the identification of miRNAs as additional biomarkers that can describe the functional characterization of cancer types. Correlation algorithms provided by GEPIA can statistically correlate survival and expression levels of genes, supporting a role for differential gene expression studies and the identification of prognostic and/or therapeutic targets.

Regarding more aggressive cancer types that lack responsiveness to first-line, mainstream therapeutic options, the identification of molecules that can give insight into expected disease progression at earlier stages is critical. For example, head and neck squamous cell carcinoma (HNSCC) presents great therapeutic challenges, as HNSCC patients are typically unresponsive to standard therapy, and disease progression surpasses surgical resection in greater than 50% of cases [64]. Thus, the identification of biomarkers as predictive factors can aid in the diagnosis and prognosis of difficult-to-treat cancers. The observation of differentially expressed genes, as stated earlier, contributes immensely to the identification of prognostic molecular markers. A study evaluated the expression patterns of the lncRNA HNF1A-AS1 in osteosarcoma patient tissues compared to adjacent non-tumor tissue. This study reported that HNF1A-AS1 is overexpressed in osteosarcoma tissue, and this increased expression is correlated with poor prognosis and decreased overall survival. Also contributing to prognostic value, within this patient cohort, high HNF1A-AS1 expression was also correlated with lung metastases. Data obtained from patient sera revealed a significantly higher HNF1A-AS1 expression in pre-operative patients when compared to expression levels in sera of post-operative patients. Availability of RNA predictive value enabled confirmation of HNF1A-AS1 osteosarcoma-derived expression as well as a feasible approach to obtaining patient samples from prognostic evaluation [65] (Cai et al., 2017).

Further, circRNA circ-ZKSCAN1 has been identified as a potential prognostic factor for bladder cancer (BCa). It was found that low circ-ZKSCAN1 expression is correlated with disease recurrence and decreased disease-free survival. Circ-ZKSCAN1 expression was higher in healthy tissue, suggesting that its downregulation in tumor tissue contributes to carcinogenic establishment, as well as a potential role for this molecule as a tumor suppressor. Mechanistically, circ-ZSKCAN1 functions as a molecular sponge for miR-1178-3p, a miRNA molecule that negatively regulates p21 expression, a critical cell cycle regulator. Thus, low circ-ZSKCAN1 results in an increase in miR-1178-3p expression and lower p21 expression, ultimately removing a critical layer of cell cycle control [66]. This premise further describes how the up- and down-regulation of ncRNA molecules result in phenotypic alterations that rely immensely on the function of their targets.

As stated, lncRNA molecules are found in bodily fluids in addition to tissue samples, making them feasible biomarker candidates. Identification from liquid biopsies makes these molecules plausible candidates, as this supports a non-invasive method of tumor profiling and assessment/monitoring of disease stage. Additionally, liquid biopsy can contribute significantly to data uniformity, which would aid in rapid identification and classification of markers across a spectrum of cancer types [67,68,69,70] (Figure 5). A “good” prognostic marker would be one that contributes to core carcinogenic patterns, such as migration, invasion, proliferation, clonogenicity, etc. Understanding the molecular mechanisms employed to contribute to these patterns gives substantial insight into the transformative capabilities of carcinogenic phenotypes.

The use of RNA molecules to characterize tumor stages has enabled significant advancement in the prognostic and diagnostic evaluation of cancer types. As mentioned, lncRNA molecules can localize as exosomal cargo. The exosomal lncRNA DANCR was evaluated for its prognostic role in BC patients. It was found that serum levels of DANCR were higher in BC patients when compared to healthy sera. Additionally, high DANCR expression in BC was directly correlated with advanced TNM staging criteria and increased LN metastasis, thus correlating with overall disease prediction. High DANCR serum levels were associated with a decrease in 5-year survival rates of BC patients and were directly correlated to ER status and HER2 status—criteria that heavily govern BC typing. Cellular mechanisms of DANCR have been associated with advanced epithelial-mesenchymal (EMT) transition and increased cancer stemness. Increased DANCR expression was also reported in ovarian cancer (OC) and was shown to promote carcinogenic mechanisms mediated by IGF2 (insulin-like growth factor 2) [71].

The use of ncRNAs as prognostic factors has been evaluated based on the differential expression patterns that exist between cancerous and non-cancerous tissue. However, a valuable tool for evaluating transcriptomic expression patterns would be an evaluation of how ncRNAs may shape a progressing tumor’s molecular profile. Further, a study was conducted that evaluated the transcriptomic expression patterns of primary BC vs. highly metastatic BC cell lines. A plethora of differentially expressed lncRNAs were identified, suggesting varied epigenetic programs and gene expression parameters of progressive disease. A novel lncRNA, lncRNA-45, was identified as the most significantly upregulated transcript in the metastatic BC cell line used. Mechanistically, lncRNA-45 was reported to increase EMT, significantly impacting the BC cell’s ability to migrate and invade. Identification of lncRNA-45 function enabled further investigation of molecular mechanisms that are employed in metastatic vs. primary BC [72]. Therefore, uniformity in tumor profiling may be accomplished at the RNA level.

## 12. Therapeutic Intervention

Advances in gene editing and understanding epigenetic factors that orchestrate gene expression have enabled the development of RNA-based therapies. This premise has given rise to the use of gene therapy as a therapeutic modality to treat disease. There are several Food and Drug Administration (FDA)-approved RNA-based therapeutic options that fall under either siRNA or antisense oligonucleotide (ASO) functional categories [73] (Figure 6). These approved RNA-based therapeutics are designed to treat pathologies such as diabetic neuropathy, hemophilia, and primary hyperoxaluria type 1 [74]. There is a broad range of therapeutics in clinical trials that are evaluating treatments using siRNA-based or ASO-based RNA therapeutics as a result of pre-clinical successes in a multitude of cancer types.

siRNAs are double-stranded noncoding RNA molecules that have similar biological functions to miRNAs but are synthetically derived (exogenously) through the process of RNA-interference (RNAi). Like miRNAs, siRNAs degrade target mRNAs in a sequence-specific manner, leading to gene silencing. Thus, the RNAi machinery targets and silences pathologic mRNA sequences. For example, an siRNA drug, siG12D-LODER (Local Drug EluteR), was developed to target KRAS, a key gene driver of uncontrolled cell signaling in cancer, in patients with locally advanced pancreatic cancer (LAPC) (Phase I; NCT01188785). This drug was directly administered to the tumor and given in combination with chemotherapy. Of note, following treatment for four months, there was a marked decrease in the tumor marker CA19-9 expression, and there was no reported tumor progression in 12 of the 15 treated patients [75]. Another clinical trial (Phase I; NCT01591356) is investigating the effects of an siRNA against EphA2 (Ephrin receptor A2) (EphA2-targeting DOPC (1,2-dioleoyl-sn-glycero-3-phosphatidylcholine)-siRNA), a protein that is associated with driving migratory and invasive patterns of tumor cells, to treat patients with several forms of advanced malignant solid neoplasm. EphA2-siRNA is administered intravenously periodically over 21-day cycles. This study is currently in progress [76].

ASOs are 12–15 nt in length and target either coding or noncoding oligonucleotides in a highly sequence-specific fashion to either block or cleave target RNAs. ASO action relies on sequence complementarity to a target molecule via traditional base pairing rules to ensure effective hybridization. This also ensures sequence specificity, eliminating the chance of off-target effects. ASOs can directly cleave the RNA target or block (via steric hindrance) an RNA molecule, both of which prevent the action of the target. Both of these actions depend on the design of the ASO. ASOs with RNase H activity function through enzymatic cleavage of RNA targets, whereas steric blockage primarily involves direct binding, serving as a “decoy”, preventing the target’s ability to exert its biological function. ASOs have the ability to biologically bind and repress regions of lncRNAs, modulating genome function effectively both in vitro and in vivo, making lncRNAs promising future targets [77].

ION-537, an ASO inhibitor of YAP1 (yes-associated protein 1), is currently being investigated in a clinical trial for the treatment of patients with molecularly selected advanced solid tumors (Phase I; NCT04659096) [78]. YAP1 is a protein associated with driving cell proliferation and bypassing apoptotic induction; therefore, targeting this protein may harness solid tumor progression. ION-537 will be administered intravenously through 28-day cycles. Another clinical trial investigated the efficacy of an ASO against IGF-R1 (Insulin-like Growth Factor), termed IGF-1R/AS ODN, for treating patients with newly diagnosed malignant glioma (Phase I; NCT02507583) [79]. This study involved autologous treatment; tumor cells were isolated from patients at the time of surgery, treated with IGF-1R/AS ODN, and re-implanted into the patient. Mechanistically, exosomes released via treated tumors will carry tumor antigens that will complement the ASO targeting and mediate the apoptotic programs of tumor cells [80]. The current trials described above further delineate the versatility of RNA therapies at various stages of cancer.

RNA therapeutic development is indebted to the identification of ncRNA mechanisms of action. Additionally, ncRNAs have given way to additional targets that can aid in therapeutic efficacy, either as standalone or combination therapy. For example, the circRNA circMED27 has been identified as both a prognostic factor for HCC and a significant contributor to therapeutic resistance. circMED27 expression was upregulated in HCC patient tissue and sera, which correlated significantly with a poor prognosis. Interestingly, the mechanistic action of this circRNA was identified as a “molecular sponge” of miR-633-3p, an miRNA molecule that negatively regulates USP28 (ubiquitin-specific peptidase 28) expression, therefore functioning as a tumor suppressor. Sponging of this miRNA was identified to significantly contribute to lenvatinib resistance through alterations in USP28-related cellular mechanisms in HCC—mechanisms that have been identified as lethal disease contributors. circMED27 competes with lenvatinib for miR-633-3p binding; therefore, circMED27 upregulation promotes these USP28-related mechanisms, which, in turn, cause therapeutic resistance in HCC cells [81]. Examples as such describe the need for the identification of molecular players that contribute to therapeutic resistance and also emphasize the putative use of these molecules in combination with well-established therapies. Perhaps these potential modalities can increase tumor amenability to therapeutic approaches. This is extremely important, especially in cancers that are relatively resistant and refractory to conventional therapies.

## 13. Obstacles for RNA Therapies

On-target specificity is a central obstacle to overcome when developing an RNA-based therapeutic that employs gene silencing. RNA is an unstable molecule; therefore, the development of ways to enhance RNA stability as well as its uptake remains prominent. Additionally, siRNA molecules are structurally susceptible to RNases and degradation. Some advances in these obstacles have begun, which employ various chemical and structural modifications that make RNAi delivery more physiologically amenable. For example, use of a GalNAc (N-acetylglucosamine)-siRNA conjugate aids in delivery and uptake into cell targets (i.e., in the liver) [82]. Lipid nanoparticles (LNPs) have been developed and have enabled successful systemic delivery of RNAi. This platform was shown using ALN-VSP02, which targets vascular endothelial growth factor (VEGF)-A and kinesin spindle protein (KSP) in liver tumors [83]. Structurally, RNA is composed of a phosphate backbone, ribose sugar, and a base. Each of these RNA components can be modified to overcome their susceptibility to degradation. Questions of stability and barriers to delivery were universally overcome due to the success of the mRNA vaccine for COVID-19 in early 2021 [84], opening many doors for targeted RNA therapies.

For ASOs, it is important to take into account the folded RNA secondary and tertiary structures, which would limit target accessibility. Several programs have been developed that can make the prediction of unfolded areas of the target sequence more “targetable” (i.e., mfold [85]). Additional software, such as MAST (mRNA accessible site tagging), can be employed to effectively map accessible sites on the target for the ASO to bind [86]. Predicting and developing an ASO with the correct sequence can significantly impact the degree of binding and target success. The aforementioned use of RNase H for site-specific cleavage aids in another obstacle due to the fact that RNase H activity itself is sequence-independent, relying strongly on the degree of GC content. This therefore eliminates any enzymatic bias, exclusively relying on the ASO, as well as a drastic increase in thermodynamic stability [87].

Other than effective genomic targeting, RNA therapeutics are futile unless they have the ability to recognize and target the tumor. Nanocarriers have aided in tumor targeting, specifically regarding their structural composition (i.e., surface charge and size of the carrier). This targeting premise is known as the enhanced permeability and retention (EPR) effect. This concept centers on the proper establishment of the nanocarrier in order to optimize tumor targeting as well as carrier retention upon administration. For example, an ideal nanoparticle size should be around 50–200 nm in diameter [88]. It is important to note that nanocarrier optimization varies depending on the area of the tumor due to permissive physiological variability. While these obstacles are currently being tackled in the literature both in vitro and in vivo, as well as in other studies in the very early stages of clinical trials, there is substantial evidence thus far supporting the future use and success of these RNA-based therapies.

## 14. Conclusions

Cancer traditionally arises from a series of genomic modifications that catalyze cellular transformation and differentiation. Genomic instability has previously been attributed to an accumulation of mutations that aid in the ability of a cell to acquire carcinogenic ability. Differential gene expression patterns that drive these processes serve as an alternate route for study and highlight the mechanisms that cancer cells employ to promote their atypical phenotypes. ncRNA molecules provide an additional layer of regulation and molecular innervation that contributes greatly to both genomic instability and carcinogenic transformation. These molecules represent the vast majority of the genome and have the ability to regulate cell activity on a genomic, transcriptomic, proteomic, and epigenomic level, surpassing, in quantity, the list of roles provided by other cellular factors. The versatility, tissue-specificity, and disease-specific modalities of ncRNAs make them attractive candidates for study, whether it is to identify novel prognostic or therapeutic options or to further understand the complexity of the intracellular mechanisms of cancer cells. A significant obstacle in cancer research is the heterogeneity that exists both between cancer types and among patients with the same cancer. Exploration of novel avenues of molecular mechanisms can provide a vastly increased array of targetable molecules that can aid in bypassing these obstacles. In the near future, we can foresee numerous additional biomarkers (either specific to a cancer or more broad) that can be evaluated in normal screenings, contributing to much earlier diagnoses. As such, biomarker discovery and evaluation may change the scope of cancer research, detection, and treatment as we currently know it. Thus, personalized medicine can be taken to a new level by utilizing “junk DNA” and bringing light to “dark matter”.

## Figures and Tables

**Figure 1 ijms-24-09914-f001:**
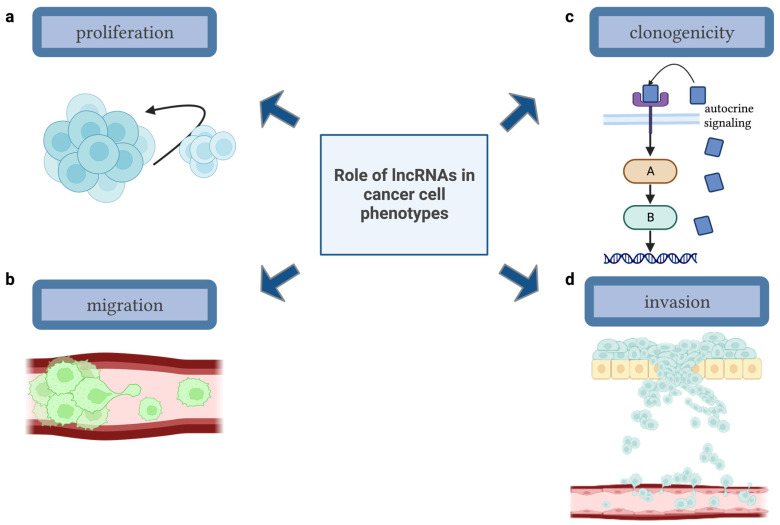
Overview of lncRNA roles in cancer cell phenotypes. lncRNA molecules modulate gene expression and protein localization in cells, thus driving central carcinogenic phenotypes such as (**a**) establishing cellular immortality; (**b**) gain-of-function or unregulated motility of cancer cells, diving cellular migration; (**c**) promoting the self-growth and sufficiency of cancer cells by driving autocrine signaling; and (**d**) contributing to the ability of a cancer cell to degrade an extracellular matrix and invade local and distant tissues, contributing to metastasis. [Figure adapted from BioRender.com].

**Figure 2 ijms-24-09914-f002:**
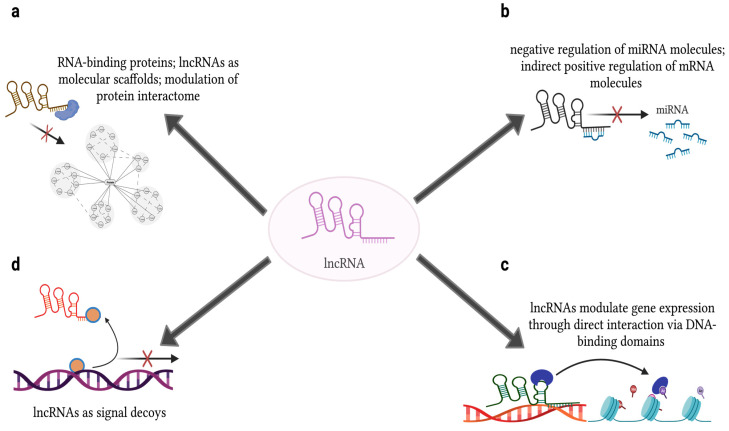
Roles of lncRNAs within the cell. lncRNAs have been mainly identified to function as (**a**) protein scaffolds via extensive protein-binding domains, thereby modulating the protein interactome; (**b**) negative regulators of miRNAs via sequence complementarity of appropriate seed length; (**c**) molecular scaffolds via DNA-binding domains; and (**d**) signals and signal decoys that directly mediate gene expression. [Figure adapted from BioRender.com].

**Figure 3 ijms-24-09914-f003:**
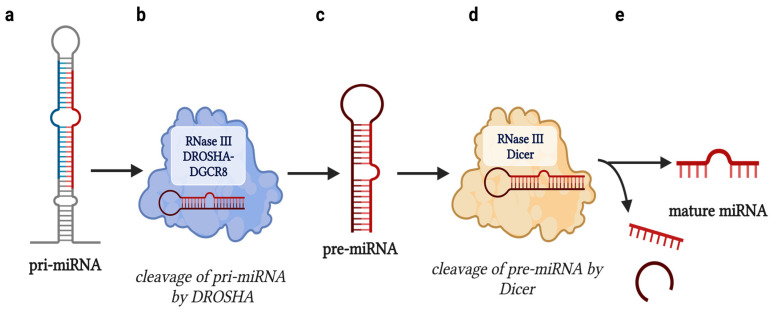
Processing of primary miRNA transcripts into mature miRNA molecules. (**a**) miRNAs are produced as pri-miRNA transcripts; (**b**) pri-miRNA transcripts are enzymatically processed via RNase III DROSHA-DCRG8 complex, producing (**c**) pre-miRNA transcripts that are (**d**) enzymatically processed into mature miRNA molecules via the RNase III Dicer; (**e**) the entire biological process yields mature miRNA transcripts that contain a 5′ terminal phosphate and a 3′ hydroxyl group, and is structurally competent for mRNA targeting. [Figure adapted from BioRender.com].

**Figure 4 ijms-24-09914-f004:**
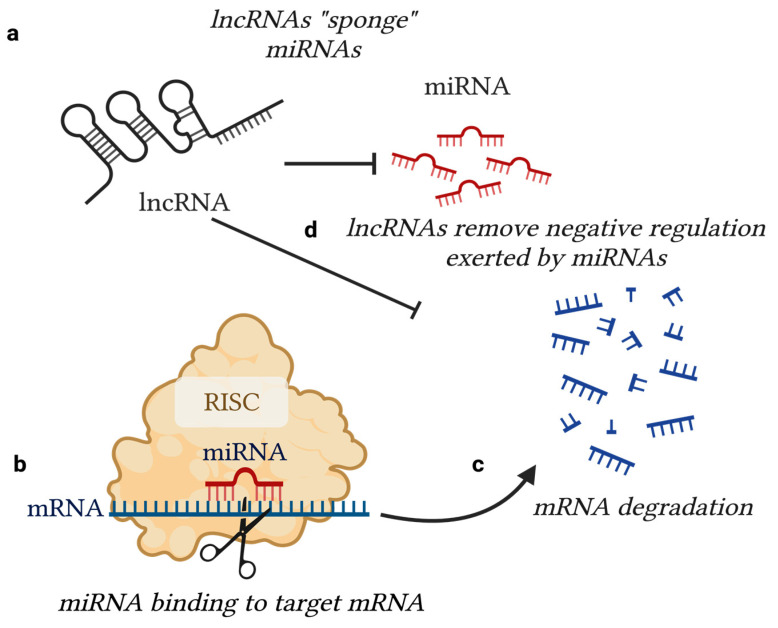
lncRNA-miRNA-mRNA regulatory axis of gene expression. (**a**) lncRNA molecules “sponge” miRNAs via sequence complementarity, serving as negative regulators of miRNA expression; (**b**) miRNAs target mRNAs in a sequence-specific fashion through interaction with a RISC-guided complex; (**c**) miRNAs bind directly to mRNAs and inhibit their translation through direct interaction with the 3′UTR of the mRNA via MREs; (**d**) lncRNAs therefore indirectly regulate mRNA translation at the miRNA level. [Figure adapted from BioRender.com].

**Figure 5 ijms-24-09914-f005:**
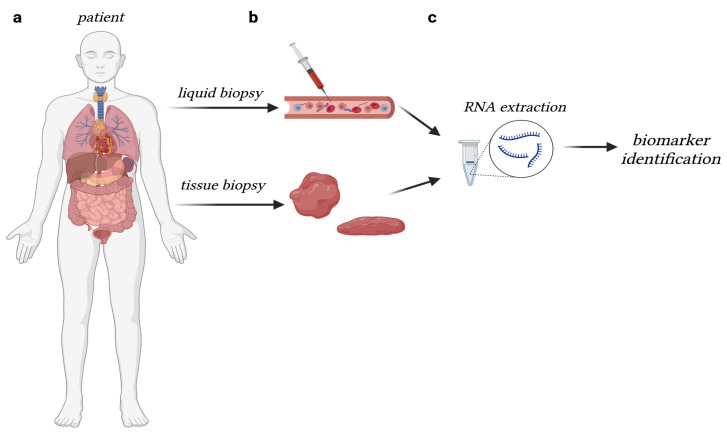
Presence of RNA molecules in liquid and tissue biopsies. (**a**) lncRNA molecules are present in body fluids and tissue samples, making them excellent candidates for detection; (**b**) RNA extraction from patient samples from multiple routes, aiding in (**c**) the rapid identification and classification of markers in a spectrum of cancer types. [Figure adapted from BioRender.com].

**Figure 6 ijms-24-09914-f006:**
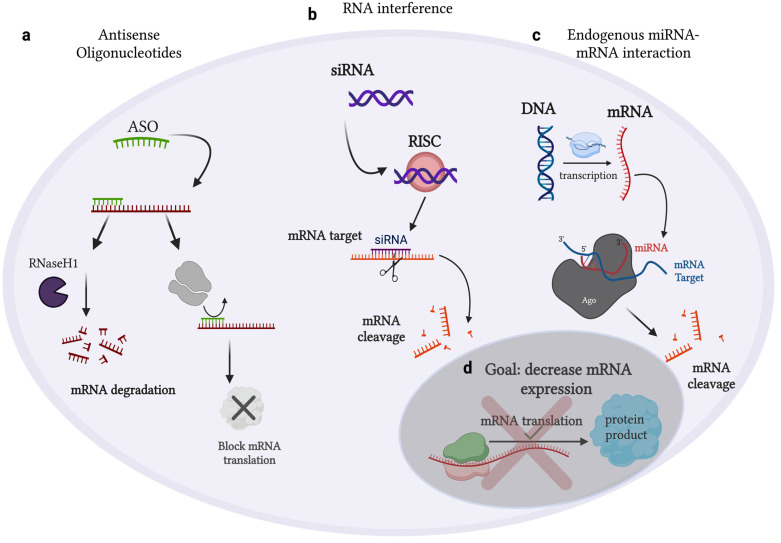
Endogenous cellular mechanisms of RNA therapeutics. (**a**) ASOs can directly cleave an RNA target or block (via steric hindrance) an RNA molecule in order to hinder expression of the mRNA target; (**b**) synthetic dsRNA molecules, known as siRNAs, exert similar biological functions as miRNAs and target mRNA molecules in a sequence-specific fashion, leading to mRNA cleavage via RISC-mediated catalysis; (**c**) physiological process of mRNA regulation via miRNA interactions; (**d**) the goal of RNA therapeutics is to target mRNAs in a highly sequence-specific fashion and block the translation of a functional protein product that has been identified to contribute to pathological conditions. [Figure adapted from BioRender.com].

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
