# Peer review of "Interactome of Long Non-Coding RNAs: Transcriptomic Expression Patterns and Shaping Cancer Cell Phenotypes"

_ijms, 2023, doi:10.3390/ijms24129914_

Round 1

Reviewer 1 Report

Difficult subject and hard to concentrate all the published data. I appreciate the effort and stratification of the article. However considering the title, my expectations were bigger in terms of oncological impact of selected item. Agree with subchapter - prognostic role. What about of future predictive role of miR?

Could you consider to discuss a little bit the advantage of RNA for better and dynamic characterization of the tumor- I am thinking for example for breast cancer, the recommendation to make biopsy at each disease progression for HER2 status? 

What about the resembles and differences between primary tumor and metastasis?

Author Response

Please see attached response under Reviewer #1. Thank you for your feedback!

Reviewer 2 Report

In this review manuscript, DeSouza et al. summarized the role of diverse non-coding RNAs in cancer, focusing on their function and expression levels. It is very impactful to have this review in the field, and I have a few comments on this manuscript:

1, The manuscript mentioned circular lncRNA, but didn’t provide further information about that. It would be helpful to discuss their function, even briefly.

2, It has been shown that piRNAs are also associated with certain types of cancer. I suggest the authors also make a summary of this topic. Also, lncRNAs can be piRNA precursors.

3, What are the challenges of discovering new functions of non-coding RNAs?

4, Any examples of good diagnostic/prognostic lncRNA markers?

5, There are many types of non-coding RNA discovered. What are the next steps and focuses in this field? How would the authors envision the near (<5 years) and long future (>10 years) of this field? Discussion this is helpful for the readers of this review.

Author Response

Please see attached document for responses to reviewer #2. Thank you for your kind and constructive feedback!
